# Apical Sperm Hook Morphology Is Linked to Sperm Swimming Performance and Sperm Aggregation in *Peromyscus* Mice

**DOI:** 10.3390/cells10092279

**Published:** 2021-09-01

**Authors:** Kristin A. Hook, Lauren M. Wilke, Heidi S. Fisher

**Affiliations:** Department of Biology, University of Maryland, College Park, MD 20815, USA; dr.kristinhook@gmail.com (K.A.H.); l_wilke0617@email.campbell.edu (L.M.W.)

**Keywords:** sperm aggregation, sperm morphology, sperm motility

## Abstract

Mammals exhibit a tremendous amount of variation in sperm morphology and despite the acknowledgement of sperm structural diversity across taxa, its functional significance remains poorly understood. Of particular interest is the sperm of rodents. While most Eutherian mammal spermatozoa are relatively simple cells with round or paddle-shaped heads, rodent sperm are often more complex and, in many species, display a striking apical hook. The function of the sperm hook remains largely unknown, but it has been hypothesized to have evolved as an adaptation to inter-male sperm competition and thus has been implicated in increased swimming efficiency or in the formation of collective sperm movements. Here we empirically test these hypotheses within a single lineage of *Peromyscus* rodents, in which closely related species naturally vary in their mating systems, sperm head shapes, and propensity to form sperm aggregates of varying sizes. We performed sperm morphological analyses as well as in vitro analyses of sperm aggregation and motility to examine whether the sperm hook (i) morphologically varies across these species and (ii) associates with sperm competition, aggregation, or motility. We demonstrate inter-specific variation in the sperm hook and then show that hook width negatively associates with sperm aggregation and sperm swimming speed, signifying that larger hooks may be a hindrance to sperm movement within this group of mice. Finally, we confirmed that the sperm hook hinders motility within a subset of *Peromyscus leucopus* mice that spontaneously produced sperm with no or highly abnormal hooks. Taken together, our findings suggest that any adaptive value of the sperm hook is likely associated with a function other than inter-male sperm competition, such as interaction with ova or cumulous cells during fertilization, or migration through the complex female reproductive tract.

## 1. Introduction

The sperm cell has undergone dramatic evolutionary diversification and extreme modifications such as loss of the flagella, gigantism, polymorphism, or even adhesion with other sperm to form conjugates, as reviewed in [1,2]. In Eutherian mammals, sperm typically feature a round, paddle-shaped and symmetrical sperm head, but several lineages of muroid rodent species have independently evolved sperm cells with a falciform apical hook—a cytoskeletal structure extending from the sperm head that is composed of acrosomal material [3,4,5,6]. A few rare species lack sperm hooks entirely [3,4,6,7,8], whereas some possess multiple sperm hooks [8,9,10,11]. For those species that do possess a sperm hook, its size, shape, and position is highly variable across species [3,5,12]. Given that sperm cells shed most of their organelles and cytoplasm during development [13], the structures retained, including morphological head features such as the apical hook, are likely to confer an adaptive function [1], and references therein. Despite the attention this unique head structure has received over the last two decades, its functional significance remains poorly understood. 

Several non-mutually exclusive functional hypotheses have been suggested for the apical sperm hook, including that it is an evolutionary response to the female environment [14]. For example, the hook is suggested to help the sperm exert force to penetrate the ova [15], or as an adaptation to the complex microenvironment of the female reproductive tract [16] by facilitating either the adherence of sperm to the epithelium of the oviductal wall during a short quiescent phase during their migration prior to fertilization [17] or promoting swimming patterns that are conducive to their motility in the chemical or physical properties of the female tract [3]. A study comparing three species of Australian rodents—two with three-hooked sperm and one without hooks—failed to find any correlation between the presence of sperm hooks and the thickness of the zona pellucida in freshly ovulated oocytes [15]. This finding, in addition to the observation that the sperm hook deploys prior to fertilization in the wood mouse (*Apodemus sylvaticcus*) [18], does not support the hypothesis that the sperm hook facilities ova penetration. Moreover, a study in hamsters (*Mesocricetus auratus*) observed that sperm attachment to the oviductal epithelium appears to play an important role in sperm viability and therefore fertilizing potential [17]. Observations of house mouse (*Mus musculus*) sperm in situ show that they indeed retain these optimal positions within the reproductive tract by adhering to the epithelium using their hook [19]. Further indirect evidence that is consistent with the hypothesis that the sperm hook facilitates sperm attachment to the oviduct comes from a comparative analysis across murine rodents, which found that estrus duration correlates positively with sperm hook curvature [20]. Furthermore, there is some indirect evidence that the apical hook helps sperm swim faster. In a comparison of closely related rodent species, those with slower sperm velocities had sperm heads without a hook and a smaller head area [7], although given that head area is in part determined by the presence of the hook, it is hard to disentangle whether the reduced velocity was driven by overall changes in the head shape itself, which is known to affect sperm hydrodynamics [21,22]. However, another comparative study in murid, crecitid, and arvicolid rodents found that hook curvature and length positively associate with sperm swimming speed [23]. 

Another hypothesized function for the apical sperm hook is that it is an adaptation to sperm competition [18], an important selective force in the evolution of sperm traits [24,25] that occurs when the sperm from two or more males compete for the fertilization of a given set of female ova [26]. Comparative studies testing this functional hypothesis of the sperm hook across species have yielded supporting evidence. For example, an investigation of the shape and curvature of the apical hook across 37 murine rodent species showed that these hook features correlate with relative testes size, a proxy for sperm competition risk [12]. Another cross-species study in the largest murine rodent tribe found that residual testis mass also correlates with the apical hook length and angle, with high levels of inter-male sperm competition resulting in a longer hook and low levels of inter-male sperm competition resulting in more divergent sperm forms with reduced or absent apical hooks [11]. Similar findings were observed for a comparison of muroid rodents [27], a smaller subset of murine rodents [28], as well as a broader group of rodents [23], the latter two of which additionally found that increased levels of sperm competition associates with reduced variation in hook length, indicating stabilizing selection on this trait. In addition, in an investigation of small murine rodents from species of three separate tribes in which males have low relative testis mass, their sperm heads were highly variable in their morphology and lacked an apical hook [29]. While findings from these inter-specific studies suggest that the apical hook associates with sperm competition, intra-specific studies have generally found no such association. For instance, an inter-population study in the house mouse found no correlation between hook curvature and sperm competition risk [20]. A follow-up study using monogamous and polygamous selection lines within house mice found no evidence of divergence in hook shape between the populations, suggesting that sperm competition does not influence the morphology of the hook within this species [30].

In conjunction with inter-male sperm competition, the apical sperm hook has been hypothesized to influence the evolution of sperm aggregation [18], a relatively rare phenomenon that occurs when two or more cells join together for motility or transport through the female reproductive tract before dissociating prior to fertilization [1,2]. A highly divergent group of rodent species that exhibit both apical sperm hooks and aggregation include the three hooked sperm of the plains mouse (*Pseudomys australis*) [31], the deer mouse (*Peromyscus maniculatus*) [32], and the wood mouse, in which cell-to-cell adhesion was observed to occur along the inner surface of the hook with either the hook or flagellum of another cell [18]. In line with this hypothesis, a theoretical model predicts that the apical hook may facilitate the formation of sperm groups [33], which increase cellular velocity in some species and thus may offer a competitive fertilization advantage [2,12,18,34,35]. In contrast, some studies found no such role for the apical hook. For example, in vitro observations of house mouse sperm showed that aggregates do not typically attach to one another using their hooks [20] and that sperm of the sandy inland mouse do not aggregate at all, despite possessing a three-hooked morphology [10]. In addition, in another cross-species study of 25 muroid species, the presence of the hook was not found to associate with the percentage of sperm in aggregations, although the majority of sperm in these species did not aggregate [27]. It remains unknown whether the hook plays a critical role in the formation of sperm aggregates for species in which these coordinated sperm groups are highly motile and frequent.

Here we examine spermatozoa from closely related species of *Peromyscus* rodents (Figure 1) with naturally varying levels of sperm competition [32], as reviewed in [36], sperm head shapes [37,38], and sperm aggregation [32,35], allowing us to test the functional hypotheses that the apical sperm hook associates with sperm competition, sperm aggregation, and sperm motility within a single lineage of rodents. First, we quantified morphological variation in the hook length, width, and area across *Peromyscus* species using high resolution images and examined correlations with relative testis size, sperm aggregate frequency, sperm aggregate size, and sperm velocity. From these analyses, we discovered a subset of males within our captive population of *P. leucopus* that produce sperm with morphologically abnormal or absent hooks. Exploiting this unique opportunity to explore the effects of an abnormal hook phenotype, we were able to corroborate our findings for the broader species group on the influence of the sperm hook on sperm velocity. While we found no association between hook morphology and relative testis size, we found that relative hook width significantly negatively associates with sperm aggregate frequency and size, as well as sperm speed. Overall, our findings provide further support that the apical hook does play a role in sperm cooperation and sperm motility.

## 2. Materials and Methods

### 2.1. Sperm Collection

We obtained captive Peromyscus maniculatus bairdii, P. polionotus subgriseus, P. leucopus, P. eremicus, and P. californicus insignis from the Peromyscus Genetic Stock Center at the University of South Carolina, and P. gossypinus gossypinus from Dr. Hopi Hoekstra at Harvard University. To control for variation in life experience, we sampled all available captive Peromyscus species and avoided wild-caught specimens. We housed all sexually mature mice in same-sex cages at 22 °C with a 16L:8D light cycle and provided food and water ad libitum. We obtained sperm samples from sexually mature males and accounted for relatedness among the focal males by assigning siblings a unique ‘Family ID’. 

To collect live sperm cells, we sacrificed males via cervical dislocation after isoflurane overdose, immediately dissected, made several incisions in the caudal epididymis, and submersed the tissue in 50–1000 μL Modified Sperm Washing Medium (MSWM; 9984, Irvine Scientific, Santa Ana, CA, USA), based on epididymal size to standardize cell densities across all males. We then agitated the tissue at 300 rpm (ThermoMixer F1.5, Eppendorf, Hamburg, Germany) at 37 °C for ten minutes, inverting the tube at the five- and ten-minute mark before incubating the tissue undisturbed for another two minutes. We then collected live sperm for analysis from just below the meniscus, to enrich for the most motile cells [39], using cut tips with a wider opening so as not to disturb already formed sperm aggregates. We continued incubating the sperm sample at 37 °C until processing. In the meantime, we used a computer-assisted sperm analysis (CASA) system (Ceros II^TM^ Animal, Hamilton Thorne, Beverly, MA, USA) to rapidly assess sperm concentrations within a 3 μL aliquot of each solution in 20 μm deep chamber slides (SC- 20-01-04-B, Leja, Nieuw-Vennep, The Netherlands) at 100 × magnification with phase contrast (Axio Lab.A1, Zeiss, Jena, Germany). We later verified that sperm density estimates from CASA significantly positively correlated with manual sperm density estimates (LM: F_1,115_ = 33.95, *p* < 0.001) using a Neubauer-improved hemocytometer (Paul Marienfeld GmbH & Co. KG, Lauda-Königshofen, Germany). We recorded five 5 s videos to capture a range of 60–80 cells per video, which we determined was an ideal concentration of cells for CASA tracking efficiency and downstream SEM morphological assessments. We diluted samples with media pre-warmed to 37 °C as needed until the target concentration was reached to ensure a standard sperm cell density across all focal males. 

### 2.2. Quantifying Sperm Morphology

To prepare specimens for scanning electron microscope (SEM) imaging, we gently pipetted suspended sperm onto a pre-warmed 12 mm glass coverslip incubated them at 37 °C for 15 min, and then fixed cells with 2.5% glutaraldehyde in 0.1 M cacodylate buffer at 4 °C overnight. Next, we washed cells three times in cacodylate rinsing buffer (98% Cacodylic Acid; 0.1 M with ddH_2_O) for 2 min per wash and then dehydrated them with a graded series of ethanol, diluted first with ddH_2_O and then with Hexamethyldisilazane (HMDS). The ddH_2_O dilution series included single washes in 50% ethanol for 5 min, 75% ethanol for 5 min, 90% ethanol for 5 min, 95% ethanol for 5 min, and then three washes in 100% ethanol for 10 min each; the HMDS dilution series included a single wash in 66.6% ethanol for 10 min and then in 50% ethanol for 10 min, followed by three washes in 100% HMDS for 10 min each. We then glued coverslips to stubs with carbon adhesive tapes, stored them overnight in a vacuum desiccator, and sputter coated them in gold/palladium (Balzers Union MED 010 Deposition System, Balzers, Liechtenstein) before examining them under a SEM (Hitachi SU-3500, Tokyo, Japan). We captured digital images of at least five sperm heads per male for each species at 15,000 × magnification and an accelerating voltage of 20 kV. We selectively imaged cells with apical sperm hooks that were fully visible (i.e., were not folded under or over the sperm head) to aid downstream morphological analysis of the hook. 

We imported SEM images into ImageJ (Version 2.0.0, 2017) to perform manual morphometrics for each sperm cell. We first used the ‘freehand selection’ tool to remove artifacts of specimen preparation, including debris and membrane stretching, around the entire cell head. We measured head length and width by drawing a line using the ‘straight line’ tool, such that the main body of the head was bisected in both directions. We measured head area by adjusting the image threshold in red pixilation (min: 40, max: 100) to three consecutive values such that the cell was fully red in color. To separate the hook from the rest of the sperm head, we used the ‘freehand selection’ tool by drawing a straight line from the base of the shelf-like space located just below the hook (“the nook”) to the outer edge of the hook. We measured the inner and outer hook lengths using the ‘segmented line’ tool and then calculated an average of the two values. We measured the hook width at the base of the hook (i.e., closest to the rest of the head) using the ‘straight line’ tool. We measured hook area using the same method and image threshold values used to estimate head area. Because head lengths, widths, and areas vary across males and species [38], we calculated relative hook length, width, and area by dividing the mean hook length, width, and area by the mean head length, width, and area, respectively [38].

Within the subset of *P. leucopus* males that produced hooks with varying abnormal phenotypes, our methods for measuring their sperm hooks varied slightly than our approach used for all *Peromyscus* species. We measured the length and area, but not the width, of the sperm hook. For males that did not produce a sperm hook, we recorded hook length as 0 μm and hook area as 0 μm^2^. For males that produced a small hook, it was difficult to distinguish where the nook ended and the base of the apical hook began. In these cases, we used the average nook length (0.52 μm) for *P. leucopus* cells with normal hooks to determine the start of the hook so that it could then be separated from the rest of the cell body for measuring its length and area. Moreover, the majority of sperm hooks with an abnormal phenotype featured a bulbous rather than tapered edge, causing the hook length to be overestimated using the aforementioned methods to determine length. To correct for this issue and more accurately measure hook length, we instead used the segmented line tool to draw a single line down the center of the hook from the base to its tip, hereafter referred to as the ‘mid-line’ length. This length was also measured for *P. leucopus* males that produced sperm hooks with a normal phenotype so that they could be compared with hook lengths for the abnormal phenotype. To measure hook area, we used the same aforementioned methods.

We excluded any measurements from our analysis in which the cell hook was blurry or broken. We measured all morphological traits three times and then averaged the values for a single estimate. Data were pooled across all images per male and across all males per species to characterize morphological traits and their relative values for males and species, respectively. 

### 2.3. Quantifying Sperm Aggregation and Motility

To quantify the proportion and number of aggregated sperm cells, we imaged and analyzed live sperm cells using CASA from the same samples used for morphometrics. To do so, we gently mixed 4 μL of suspended sperm, collected from the same region of the sperm supernatant and concurrent to the morphometric and hemocytometer collections, using a reverse pipetting into 12 μL of pre-warmed media on a plastic slide within a 0.12 mm imaging spacer (GBL654008, Grace Bio-Labs SecureSeal^TM^, Bend, OR, USA) that we then covered with a pre-warmed plastic cover slip. We recorded a minimum of five 5 s videos at 60 frames/s per male, except in one case in which we were only able to record four videos. The mean (±SE) duration of time between tissue dissection and video observations (i.e., ‘post-harvest time’) was 58 (±1) minutes, which varied slightly depending on how long it took us to properly dilute each sample. 

From the sperm videos obtained using the CASA system, we manually counted the number of cells associated with each recorded track, verifying each on at least three different frames per track. When tracking sperm, CASA automatically excluded any tracks in which cells left the field within the first 10 frames or entered the field or started moving after the first 10 frames. Through our manual evaluation of each track, we further removed tracks if they involved cells that were attached to debris or the slide surface, too out of focus to count the number of cells, interacted with other cells or aggregates (e.g., colliding, joining, separating, or impeding movement), or had a distance average path <30 μm (often repeats of longer, more informative tracks). From the remaining tracks, we then calculated the proportion of cells in aggregate (i.e., frequency) by dividing the total number of cells in aggregate by the total number of cells observed. We calculated the mean number of cells in aggregate (i.e., size) by dividing the sum of cells in aggregate by the total number of sperm aggregates. Last, from this dataset we removed instances where cells were immotile (i.e., unmoving) or poorly tracked by CASA (i.e., had a low number of detection points, points that jumped suddenly, or picked up parts of the flagella) to create a dataset of only motile sperm tracks to determine the curvilinear velocity (VCL) of sperm cells. We found that VCL was significantly correlated with straight line velocity across all males (R^2^ = 0.735, *p* < 0.001), so here we report only VCL for simplicity. These data were pooled either across all tracks per male or across all males per species to characterize aggregation by males and species, respectively.

### 2.4. Statistical Analyses

To investigate differences in the sperm hook morphology across our focal *Peromyscus* species, we used mean raw values for each male and separate linear models (LM) for the sperm hook length, width, and area. In addition to ‘species’, we included head length, width, and area, respectively, as separate predictors in these models rather than relative values [40,41] to control for variation in these traits among species [38]. We initially used linear mixed models (LMM) with ‘Family ID’ as a random factor, but in all cases this term did not contribute to variation in the response variable, so we reverted to using a LM. We also used LMs to examine the correlations between head and hook traits. 

To investigate the relationship between sperm behavior and sperm hook morphology, we used mean raw values for each species and separate phylogenetic generalized least squares (PGLS) regressions [42,43] to account for the evolutionary relationships among the focal species used in this study and statistically control for phylogeny. These regressions included an ultrametric phylogenetic tree of *Peromyscus*, provided by Dr. Roy Neal Platt II (Texas Biomedical Research Institute) as a covariance matrix; the phylogeny was based on sequence variation in the mitochondrial gene, cytochrome B, and generated species relationships similar to other previously established *Peromyscus* phylogenies [36,44]. Regressions included either the mean proportion of cells in aggregate, the mean number of cells in aggregate, or the mean curvilinear velocity of cells as response variables. We considered the total number of sperm cells as well as the sperm head feature that corresponded to the main predictor (i.e., sperm head length, width, or area for sperm hook length, width, or area, respectively) as explanatory variables within these regressions. One individual was noted to have an extreme aggregate size for its species, which clearly represented an outlier and which we again verified subsequently; our results were the same regardless of whether or not we excluded this individual, so we report values with this individual included in our dataset. Last, we conducted post-hoc analyses in which we included the sperm curvilinear velocity as an explanatory variable in the final model to control for sperm speed and examine the effect of the sperm hook width. We additionally investigated the relationship between sperm hook morphology and relative testis size across species using separate PGLS regression analyses, with hook length, width, or area as the response variables and testis and body mass as separate predictor variables. For all models, we used AIC model selection to determine the minimal adequate model. We additionally investigated the relationships between sperm velocity and the frequency as well as the size of sperm aggregates using LMs. Only the best fitting models are reported here. 

To investigate the sperm hook morphology within *Peromyscus leucopus*, we used mean raw values for each male and separate models for the sperm hook length and area. In addition to sperm type (i.e., normal or abnormal), we included head length and area, respectively, as separate predictors in these models. For hook length, we used a LMM with ‘Family ID’ as a random factor, but for hook area this random factor did not contribute to variation in the response and so we used a LM. We also used LMs to examine correlations between head and hook traits. In addition to analyzing absolute raw values for hook length and area within this species, we performed a principal component analysis (PCA) of the sperm hook size using these two morphological measures. PC1 of the two hook traits explained 99.2% of the variation, whereas PC2 explained 0.8% of the variation (loadings: hook area −0.71, hook mid-line length −0.71; Figure 5B). We used the scores of the first principal component (PC1) as an explanatory variable in a LMM for sperm velocity. We also considered head length, head area, and total sperm cells as predictors in this model. After model selection using ANOVA to compare models, the only predictors were PC1 and total sperm cells, with ‘Family ID’ as a random factor. Last, we conducted post-hoc LMs to determine if the testis size or sperm production differed between males who produced normal or abnormal sperm hooks. We considered male age and whether males had been previously paired with a female as covariates in these models, which we compared using analyses of variance. 

We conducted all statistical analyses in R version 3.1.2 (R Development Core Team, 2016) and created all figures using the “ggplot2” package [45] and “ellipse” package [46] with R. We visually inspected diagnostic plots (qqplots and plots of the distribution of the residuals against fitted values) to validate models. Linear mixed models were conducted using the lmer function from the “lme4” R Package [47]. PGLS regressions were performed using the “caper” [48] and “APE” [49] packages in R. Post-hoc pairwise comparisons were made using Tukey HSD adjustments for multiple comparisons using the “LSmeans” R package [50]. 

## 3. Results

### 3.1. Characterizing Sperm Hook Shape across Peromyscus

Within the six focal species of *Peromyscus* mice used in this study, we found the hook is an allometric feature of the sperm head that significantly associates with other sperm head features; cells with longer and wider hooks or with a larger hook surface area tended to have longer (LM: F_1,135_ = 108.5, *p* < 0.001) and wider (LM: F_1,135_ = 104.9, *p* < 0.001) sperm heads or a larger head surface area (LM: F_1,135_ = 239.5, *p* < 0.001), respectively. Regression coefficients between hook morphological traits revealed significant positive relationships between its length and width (LM: F_1,135_ = 86.59, *p* < 0.001), length and area (LM: F_1,135_ = 423.5, *p* < 0.001), as well as the width and area (LM: F_1,135_ = 135.2, *p* < 0.001). 

We found significant intra- and inter-species variation in the hook length, hook width, and hook area relative to the head length, head width, and head area, respectively, (Table 1, Figure 2). Controlling for head length, apical hook length significantly differed across species (Table 2, Figure 2A), and pairwise comparisons adjusted for multiple comparisons using LSmeans revealed that *P. eremicus* produce the longest relative apical hooks (*P* < 0.05 for all pairwise comparisons), whereas *P. maniculatus*, *P. leucopus*, and *P. californicus* produce the shortest relative apical hooks (*p* > 0.05 for these pairwise comparisons). Controlling for head width, apical hook width significantly differed across species (Table 2, Figure 2B), and pairwise comparisons revealed that *P. gossypinus*, *P. leucopus*, and *P. eremicus* produce the widest relative apical hooks (*p* > 0.05 for these pairwise comparisons), the latter of which also produces a hook width that is statistically similar to *P. californicus* (*p* = 0.07). Pairwise comparisons also show that *P. gossypinus* and *P. polionotus* produce similarly wide apical hooks (*p* = 0.46). Controlling for head area, apical hook area significantly differed across species (Table 1, Figure 2C), and pairwise comparisons revealed that *P. maniculatus* has an apical hook with the smallest area (*p* < 0.05 for all pairwise comparisons) and that *P. eremicus* produces a significantly larger hook area than *P. polionotus* (*p* = 0.012). Otherwise, all other pairwise comparisons for this feature were not statistically different.

### 3.2. Associating Sperm Hook Shape with Sperm Behavior across Peromyscus

While controlling for phylogenetic relationships and head width within PGLS analyses, we found that the sperm hook width significantly associates with both the frequency (F_3,2_ = 56.29, R^2^ = 0.97, *p* = 0.014) and the size (F_3,2_ = 1979, R^2^ = 1.00, *p* < 0.001) of sperm aggregates across the six focal species of *Peromyscus* (Table 2, Figure 3). Moreover, we found that relative sperm hook width significantly associates with sperm swimming speed (Table 2, Figure 4). However, none of these relationships were observed for either the relative hook length or relative hook area across these species(Table 2). There was a significant positive association between mean sperm swimming speed and the mean proportion of sperm cells in aggregate (LM: F_1,4_ = 23.38, *p* = 0.008) and mean number of sperm cells in aggregate (LM: F_1,4_ = 29.21, *p* = 0.006) for these species. When controlling for sperm swimming speed within our PGLS analyses, hook width no longer significantly associated with the proportion of cells in aggregate (PGLS: F_4,1_ = 25.31, *R*^2^ = 0.95, *p* = 0.259), but it did for the number of cells in aggregate (PGLS: F_4,1_ = 736.7, *R*^2^ = 1.00, *p* = 0.040). Last, we found no significant association between the relative testis size and hook length (F_3,2_ = 3.378, R^2^ = 0.59, *p* = 0.753), width (F_3,2_ = 5.205, R^2^ = 0.72, *p* = 0.256), or area (F_3,2_ = 2.313, R^2^ = 0.44, *p* = 0.634).

### 3.3. Comparing Normal and Abnormal Sperm Hooks in Peromyscus Leucopus

Within our subset of *P. leucopus* males, we also found that the hook is an allometric feature of the sperm head that associates with other sperm head features. Sperm cells with a longer hook or with a larger hook surface area tended to have longer heads (LM: F_1,33_ = 3.935, *p* = 0.06) and larger head surface areas (LM: F_1,33_ = 26.26, *p* < 0.001), respectively. We also found these two hook traits themselves were positively associated (LM: F_1,33_ = 967.5, *p* < 0.001). When we reduced hook length and hook area into principal components, two discrete categories of sperm hook phenotypes emerged (Figure 5A,B). Controlling for head length, the mean (±SE) hook length for cells with an ‘abnormal’ hook phenotype (0.73 ± 0.11μm) was significantly shorter than cells with a ‘normal’ hook phenotype (Appendix A; LMM: *n* = 35, t = −32.52, *p* < 0.001). For cells with the abnormal hook phenotype, the relative hook length was 0.16. Similarly, controlling for head area, the mean (±SE) hook area for cells with an abnormal hook phenotype (0.19 ± 0.01 μm) was significantly smaller than cells with a normal hook phenotype (Appendix A; LM: F_2,32_ = 1353, *p* < 0.001). For cells with the abnormal hook phenotype, the relative hook area was 0.02. We found these phenotypes significantly differed in their swimming speed; sperm with abnormal hook morphologies (PC1) had significantly greater curvilinear velocities compared to sperm with a normal hook phenotype (LMM: *n* = 30, t = 2.105, *p* = 0.0478; Figure 5C). Post-hoc analyses showed that these males did not differ in their testes size (LM: F_3,22_ = 6.516, *p* = 0.122), but males with a normal hook phenotype produced significantly more sperm (LM: F_1,14_= 7.065, *p* = 0.019) as well as a significantly greater proportion of motile sperm (LM: F_1,30_= 22.35, *p* < 0.001). 

**Figure 5 cells-10-02279-f005:**
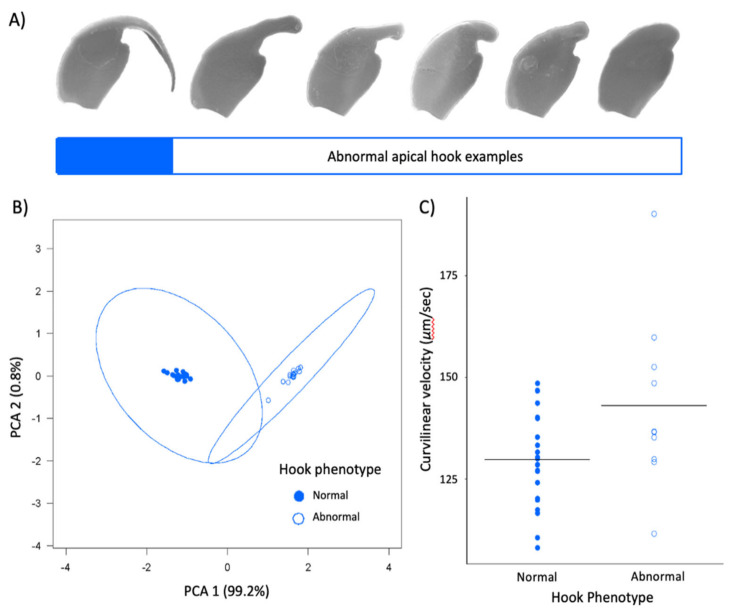
Apical hook variation in *P. leucopus* colony. (**A**) Example scanning electron micrographs of sperm with normal hooks (solid blue) and range of abnormal hook phenotypes (white). (**B**) When hook length and area are reduced to principal components, two discrete categories of males are formed based. Dots denote individual males, circles represent 95% confidence intervals. (**C**) Sperm with normal hooks (blue) are slower than those with abnormal hooks (white). Circles represent mean value for each male, and black lines represent group mean. Note truncated y-axes.

## 4. Discussion

The aim of this study was to test several hypotheses about the role of the apical sperm hook found on the sperm heads of most muroid rodents, including that it is an adaptation to sperm competition risk and aids in sperm aggregation and motility. Using a cross-species comparison of *Peromyscus* mice, we found that the hook is an allometric feature of the sperm head—larger sperm heads have larger sperm hooks. However, even when we statistically controlled for features of the sperm head, we found that the apical hook length, width, and area significantly varied across a small subset of closely related species within this genus. Despite varying in their mating systems, we found no demonstrated effect of sperm competition on sperm hook morphology. Moreover, although we did find an association between sperm hook morphology with both sperm aggregation and motility, we found that larger hooks are a hindrance rather than a helpful trait; species that produce sperm with wider hooks relative to the width of the cell’s head formed significantly fewer and smaller sperm aggregates. We also found that sperm with relatively wider hooks are significantly slower, which we verified within a subset of *P. leucopus* males that produced sperm with reduced or absent apical hooks. Post-hoc analyses for all species reveal that the effect of the sperm hook on speed explains its impact on the frequency of sperm aggregation but not aggregate size, indicating that the apical sperm hook associates with how often cells adhere to one another. Together our results suggest evolutionary costs for having an apical hook, with any adaptive benefits remaining unclear.

Post-copulatory sexual selection is an important driver of sperm diversification [25] and the apical hook is proposed to be a consequence of evolutionary modifications to the sperm head driven by sperm competition [18,23]. Despite empirical evidence in other rodent species that sperm competition influences various aspects of the hook phenotype [11,12,23,27,28,29], we did not find evidence that post-copulatory sexual selection acts on this variation in hook phenotypes present across focal *Peromyscus* mice. While it is possible that a broader sampling of species in this study might have yielded different results, it is interesting to note that studies that have found positive results regarding these associations were inter-specific comparisons across disparate rodent groups, whereas intra-specific studies have not. For example, evidence suggests that there is no association between hook curvature and sperm competition risk in natural house mouse populations [20], or those involved in an experimental evolution study [30], suggesting that sperm competition does not influence the morphology of the hook within this species. Our findings corroborate these results and suggest that this putative sperm adaptation may not be subjected to sexual selection pressure in all rodents. 

The sperm hook is believed to have evolved to facilitate the formation of sperm trains or groups that enhance a male’s competitive ability in sperm competition [18], but our results do not provide direct support for this hypothesis. We found that *Peromyscus* species that produce sperm with relatively wider, and therefore larger, apical hooks formed significantly fewer and smaller sperm aggregates. In the wood mouse, sperm form groups of cells that attach to one another through cell-to-cell adhesion along the inner surface of the hook to either the hook or flagella of other cells [18]. In this species, it seems reasonable that a longer or larger hook would scale to a larger adhesive region, and therefore a greater likelihood of forming more and perhaps larger aggregates. Within *Peromyscus* sperm aggregates, the majority of aggregated cells attach at the heads but not necessarily by latching on to one another at their hook region [34], similar to what is observed within house mice [20]. This could explain why our results show that the hook is more of a hindrance to sperm aggregation within *Peromyscus*. Other studies that have failed to find a relationship between the apical sperm hook morphology and sperm aggregation have compared them within species in which their sperm do not frequently aggregate or form few motile aggregates, e.g., [20,27]. It is also important to note that *Peromyscus* sperm aggregates observed in vitro are common and highly motile, depending on the species [35]. Despite these differences, our findings suggest that a smaller hook associates with significantly more and larger sperm aggregates. 

We also found that sperm cells with relatively wider hooks are significantly slower than sperm with narrower hooks, suggesting that, in these mice, the hook hinders rather than enhances sperm motility. Our within-species analysis of *Peromyscus leucopus* males that produced sperm with an abnormal hook phenotype corroborated these results; sperm cells with reduced or no hooks were significantly faster than sperm with a fully intact hook. This result is contrary to those from other cross-species comparisons of rodents. For instance, a study that used geometric morphometrics to examine the hook curvature angles, length, and overall shape in murid, crecitid, and arvicolid rodents found that sperm cells with a longer hook swam faster [23]. The authors attribute this finding to the particular pattern of movement exhibited by muroid rodent sperm—a “hatchet-like motion” in which the cell cuts through the fluid surrounding it with the rostral portion of its hook—and a design that resembles the bow of a boat that confers faster and more hydrodynamically efficient swimming patterns [23]. Similarly, a comparison of closely related rodent species found that those with slower sperm velocities had sperm heads without a hook and a smaller head area [7]. Given that head area is in part determined by the presence of the hook, it is difficult to disentangle whether the reduced velocity is driven by overall changes in the head shape itself, which is known to influence the hydrodynamics and therefore the swimming performance of sperm [21,22]. Within our study, we controlled for differences in head shape and still found a significant negative effect on sperm swimming speed yet were unable to control for flagellum or midpiece size, and various physiological factors, that are also likely to also influence motility. Furthermore, when we performed post-hoc analyses controlling for sperm speed, we no longer saw an association between the apical hook and the frequency of aggregation, suggesting that the speed of sperm cells impacts their encounter rate with one another, driving them to form aggregates. However, when we controlled for sperm speed within our analysis of sperm aggregate size, its significant relationship with the apical sperm hook width persisted, suggesting that the shape of the hook, not the speed of the cells themselves, influences the number of cells that aggregate across these species.

One fascinating finding from our study was a subset of *Peromyscus leucopus* males that produce sperm cells with reduced or entirely absent sperm hooks. Only 12% of males within our captive colony did so, and we exploited this variation to assess the role of the apical hook on sperm motility. Since this species does not generally form sperm aggregates [35], we were not able to assess the impact of an abnormal hook phenotype on sperm aggregation. It is important to note that these males never simultaneously produced a mix of both normal and abnormal sperm hook phenotypes, suggesting a developmental issue in the epididymis, where the hooks form post-meiotically [1,51], led to these abnormal phenotypes. We verified that relative testes size did not differ in these males between males with normal and abnormal sperm, but we found that males that produced sperm with abnormal hook phenotypes had significantly lower caudal epididymal sperm counts, suggesting reduced sperm production efficiency. An examination of sperm within Australian Hopping mice revealed a shortening or absence of the apical hook and simplification of the sperm head morphology within two species, which the authors considered to be a recently derived trait and suggest it may be due to a monogamous mating system due to relaxed levels of sperm competition [8]. Importantly, discovering the subset of males in our study with abnormally hooked or hookless sperm, we confirmed that several of these males who had been paired with a female had produced litters. Therefore, an abnormal apical hook does not necessarily prohibit sperm function in these mice. A previous rodent study of sperm lacking an apical hook but featuring a very large acrosome led the authors to conclude that, rather than the hook, these sperm rely more on enzymatic digestion of the extracellular matrix to facilitate zona penetration at fertilization [29]. This finding can help explain how species or males that lack an apical hook are still fertile, if the hook does indeed aid in fertilization [51]. 

## 5. Conclusions

In this study, we compared morphological variation in the apical sperm hook within and between species of *Peromyscus* mice and investigated the association with sperm motility and aggregation behavior. We found significant variation in hook size across species and a negative association between relative hook width with both sperm aggregation and swimming speed. Our data revealed no signature of sexual selection on hook phenotypes across these species, however. An important caveat is that our findings are based on a limited number of species and in vitro observations, the latter of which allowed us to control the environment when assessing sperm aggregation and motility but does not necessarily give us an indication of what is happening in nature within the female reproductive tract [14]. Taken together, our findings suggest that any adaptive value of the sperm hook is likely associated with a function other than inter-male sperm competition, such as interaction with ova or cumulous cells, or migration through the complex female reproductive tract. Although testing these hypotheses was beyond the scope of our study, future in vivo studies are warranted for unraveling the elusive adaptive benefit of the apical sperm hook.

## Figures and Tables

**Figure 1 cells-10-02279-f001:**
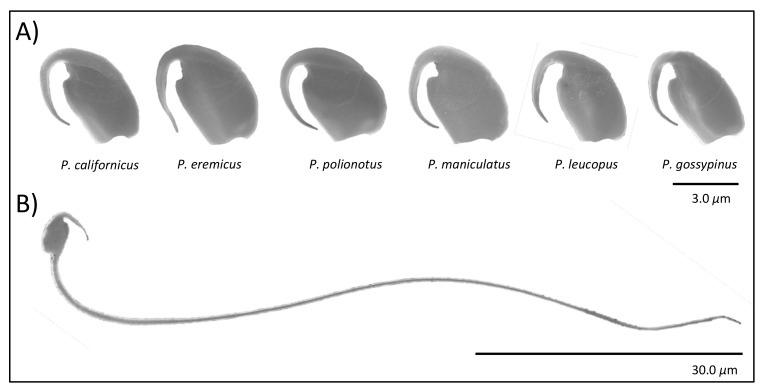
Representative images of sperm heads of six closely related species of *Peromyscus* mice (**A**) and a full *Peromyscus maniculatus* sperm cell (**B**).

**Figure 2 cells-10-02279-f002:**
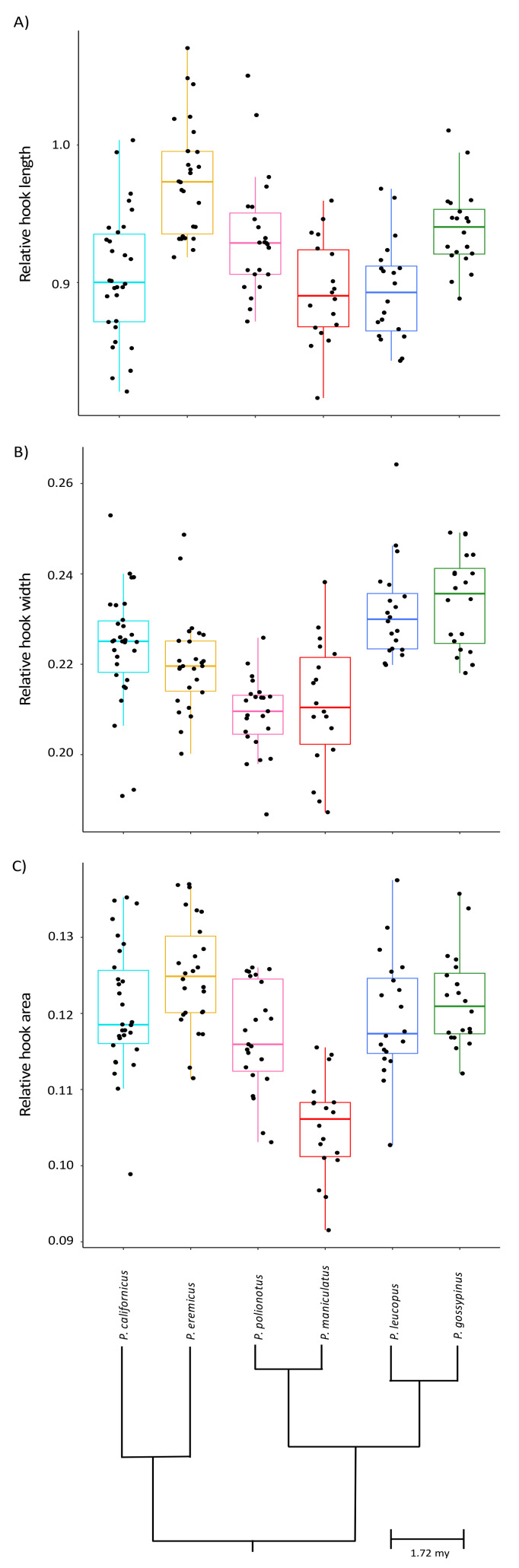
Intra- and inter-species variation in the relative length (**A**), width (**B**), and area (**C**) of the apical hook featured on the heads of sperm cells produced by six closely related species of *Peromyscus* mice. Values are relative to the length, width, and area of the sperm head, respectively. Boxplots represent median and interquartile ranges with mean values per male overlaid as black dots. Species relationships are indicated within the phylogeny. Note truncated y-axes.

**Figure 3 cells-10-02279-f003:**
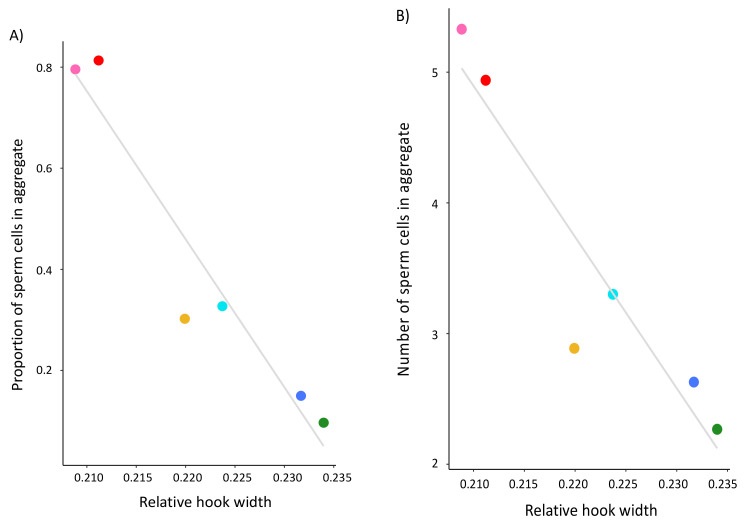
Mean width of the apical sperm hook negatively associates with the frequency (**A**) and size (**B**) of sperm aggregates produced within six species of *Peromyscus* mice. Statistical analyses were conducted using a PGLS to control for phylogenetic relationships with the regressions, which are indicated by a line in gray. In both panels, species are indicated by the same color patterns from Figure 2. Note truncated y-axes.

**Figure 4 cells-10-02279-f004:**
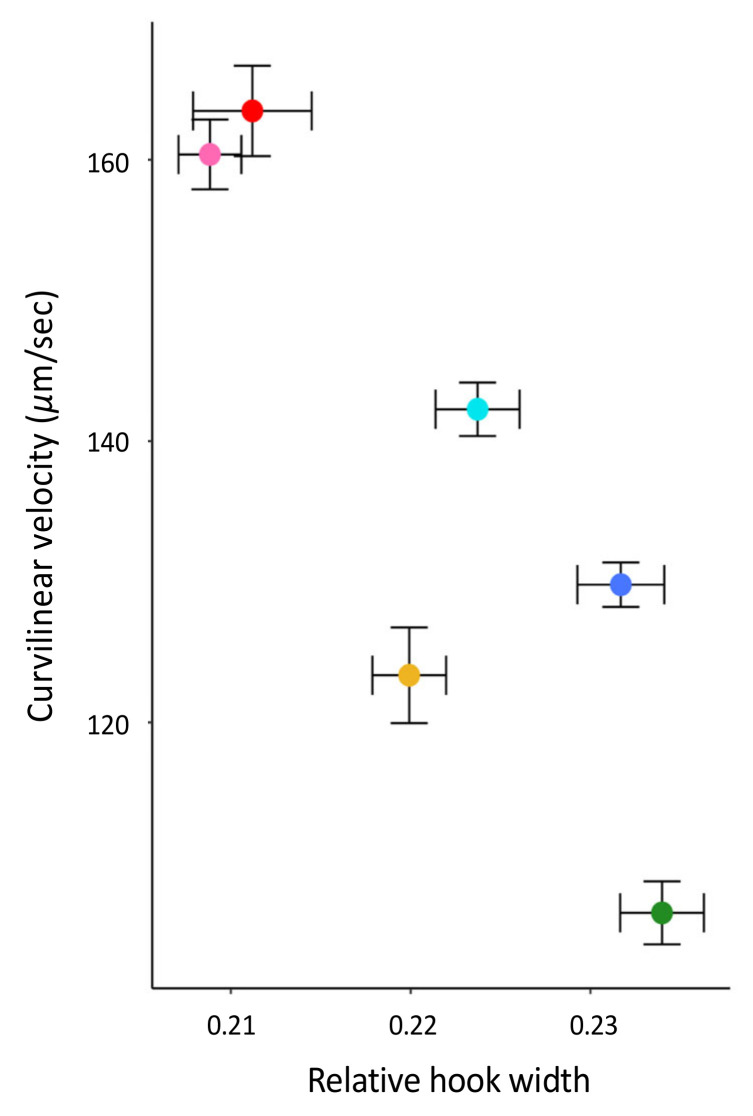
Association between relative apical sperm hook width and sperm velocity by species. Standard error bars for both traits are shown in black, and dots denote species’ mean with same colors as Figure 2. Note truncated *y*-axis.

**Table 1 cells-10-02279-t001:** Fixed effects from linear models examining differences in sperm hook morphology across six *Peromyscus* species.

Response	Model Term	Estimate (SE)	t	Pr(>|t|)
Hook length	Intercept	1.37 (0.46)		
	head length	0.60 (0.10)	6.20	**<0.001**
	*P. eremicus* *P. gossypinus* *P. californicus* *P. leucopus* *P. polionotus*	0.46 (0.06) 0.21 (0.06) 0.04 (0.05) −0.04 (0.06) 0.13 (0.06)	7.88 3.55 0.79 −0.67 2.16	**<0.001** **<0.001** 0.432 0.505 **0.033**
Hook width	Intercept	0.37 (0.10)		
	head width *P. eremicus* *P. gossypinus* *P. californicus* *P. leucopus* *P. polionotus*	0.10 (0.03) 0.03 (0.01) 0.01 (0.02) −0.02 (0.02) 0.00 (0.02) −0.02 (0.01)	2.93 3.47 0.53 −0.97 0.04 −1.73	**<0.01** **<0.001** 0.594 0.332 0.971 0.086
Hook area	Intercept	−0.08 (0.20)		
	head area *P. eremicus* *P. gossypinus* *P. californicus* *P. leucopus* *P. polionotus*	0.11 (0.01) 0.29 (0.04) 0.21 (0.04) 0.20 (0.04) 0.18 (0.04) 0.16 (0.03)	7.47 8.17 4.91 5.50 4.19 5.13	**<0.001** **<0.001** **<0.001** **<0.001** **<0.001** **<0.001**

For each model, all rows are compared with the intercept—*P. maniculatus*. Significant results at the *p* < 0.05 level are bolded.

**Table 2 cells-10-02279-t002:** Results from phylogenetic generalized least-squares regression models explaining sperm behavior-based hook morphology.

Response	Predictor	*t*	*P*	*R^2^*	λ
Proportion of cells in aggregate	Hook length	−0.355	0.741	−0.21	1.000 (0.15, 1)
	Hook width	−8.351	**0.014**	0.97	1.000 (1, 0.01)
	Head width	12.613	**0.006**		
	Total sperm cells	−2.193	0.160		
	Hook area	−2.300	0.105	0.51	1.000 (1, 0.11)
	Head area	2.680	0.075		
Number of cells in aggregate	Hook length	−0.575	0.596	−0.15	1.000 (0.18, 1)
	Hook width	−53.746	**<0.001**	1.00	1.000 (1, 0.01)
	Head width	76.135	**<0.001**		
	Total sperm cells	−12.787	**<0.01**		
	Hook area	−1.852	0.161	0.31	1.000 (1, 0.10)
	Head area	2.047	0.133		
Sperm curvilinear velocity	Hook length	−1.670	0.170	0.26	1.000 (0.19, 1)
	Hook area	−2.809	0.107	0.56	1.000 (1, 0.09)
	Head area	2.996	0.096		
	Total sperm cells	1.606	0.249		
	Hook width	−4.607	**0.019**	0.84	1.000 (1, 0.01)
	Head width	5.312	**0.013**		

In all models, the branch length transformations for lambda, λ, were set using maximum likelihood (‘ML’), with lower and upper boundaries for the lambda estimation indicated within parentheses. Stepwise model simplification comparing models with and without total sperm cells (density) did not change any of these results (based on AIC values). Reported *R^2^* values are adjusted values. Significant results at the *p* < 0.05 level are bolded, shaded areas signify different models.

## Data Availability

Data are available from Dryad.

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
