# Peer review of "Apical Sperm Hook Morphology Is Linked to Sperm Swimming Performance and Sperm Aggregation in Peromyscus Mice"

_cells, 2021, doi:10.3390/cells10092279_

Round 1

Reviewer 1 Report

This manuscript explores links between sperm head morphology (including apical hook), sperm aggregation and sperm velocity in six species of Peromyscus mice. The authors found significant variation in sperm head morphology between these species and a negative relationship between sperm hook width and both sperm aggregation and sperm velocity. They also took advantage of some males of one of these species that happened to produce sperm with a stunted or even missing hook, finding that here, too, sperm velocity was reduced in sperm with an intact hook.

Overall, this paper is an interesting addition to a series of studies on this general topic in diverse murine taxa, and it is generally well-written. However, apart from a few minor details, there are some more serious analytical issues that prevent this manuscript from being acceptable for publication in its current form. I also think that given the severe limitations of this study (primarily the comparative parts), the results should be interpreted carefully, the caveats acknowledged, and the discussion/conclusions toned down accordingly.

Major issues:

  1. I would be more cautious in dismissing any role of postcopulatory sexual selection, in that the comparative analyses exploring this role are done across only 6 species and with multiple predictors in the model, leaving 2 or even just 1 residual df. These analyses do not have enough statistical power to confidently conclude anything. In fact, a common recommendation is to have at least 10 times more observations than estimated terms in the model (e.g. see Mundry's chapter in Garamszegi 2014, Modern Phylogenetic Comparative Methods and Their Application in Evolutionary Biology, DOI: 10.1007/978-3-662-43550-2_6, Springer). These analyses of the present manuscript are nowhere near this recommendation, and I am very skeptical of any of the multiple regression analyses across these 6 species.

  2. Related to these issues, combining multiple predictors that are highly correlated among one another increases the risk of multicollinearity, which can generate false positive/negative results. It would be essential to test for collinearity (e.g. variance inflation) and, if it is a problem, to address it adequately. It is also not not clear if and how variables were used as raw data or scaled/transformed in any way.
  3. The authors focus solely on sperm head morphology. While this is justified for most analyses, including those on sperm aggregation, it seems less justified when testing for links between sperm morphology and sperm velocity. Sperm velocity is the result of thrust generated by the flagellum, energy supplied by the midpiece and drag caused by the sperm head. Thus, ignoring any variation in flagellum length in the conclusion that the sperm hook slows sperm swimming is misleading as slower swimming can result from lower propulsive forces as much as more drag.

  4. It is also unclear how sperm velocity measurements were influenced by sperm aggregation. To assess the relevance of hook width in determining swimming speed, individual sperm (not aggregates) should be compared. The contribution of sperm aggregates is a different question. So, I am wondering how the velocity of individual sperm compares between species, or how individual and aggregated sperm compare within males/species. This would be comparing apples with apples. Finally, how does curvilinear sperm velocity compare with, say, straight-line velocity? Is it possible that sperm with a thinner hook have a longer track overall but, ultimately, their directional velocity is less different from species with a wider hook (e.g. due to differential lateral head movements)?

Minor issues:

57          Please provide scientific species names at first mention.

74-91    I think there is yet another comparative study on Australasian murines (McLennan et al. 2017; Reprod. Fertil. Dev. 29:921-930) with results along these lines, in case you wish to expand the taxonomic breadth.

91-97    This is not necessarily surprising. First, there is obviously far more variation in sperm morphology and sperm competition levels across compared to within species, making it easier to find an association. Second, these between- and within-species results are not necessarily contradictory, because in principle, even multiple negative intra-specific associations could generate a positive interspecific relationship as long as species means are compared with one another (as typically done).

111-112              This sentence sounds confusing because it implies that aggregates have hooks but do not attach to one another. You probably mean …”in vitro observations showed that house mice sperm…”

123        This switch in tense is confusing, especially because sperm traits clearly had to be quantified before conducting any analysis.

197        What exactly do you mean by “top of the hook”? Say, in Fig. 4A, first sperm head, would the “top” be up or left from the “nook”? This obviously depends on the orientation. It might help to provide a schematic or describe in terms of perpendicular/parallel to head width (or head length, whichever you prefer), or terminology such as anterior/ventral/distal/rostral etc.

199        Same for hook width – it’s not entirely clear what the landmarks were: width of base (i.e. contact zone of head and hook), distance between outermost points of curvature (e.g. basically the diameter of the hook), or anything else? Also, I assume you did not measure the hook angle, which seems to be measured in many other studies on the apical hook evolution?

215/299              sperm length -> do you mean hook length?

223-224              What exactly do you mean by pooling across all males per species? I assume (and hope) this was the grand mean of all mean values per male, and not across all images treated as independent datapoints contributing to the species mean? I think “and” instead of “or” would actually be clearer.
Also, further up you mentioned that you measured at least 5 sperm per male, but what was the mean and SD? Especially for the analysis of abnormal hooks, I think a substantially larger sample size would be needed to obtain adequately robust data. I assume that there was considerable variation within each of these males as to the extent and frequency of these abnormalities across their sperm.

259-266              Since you used mixed models already, did you try setting up a hierarchical model with species > (family >) male > sperm using means within male? This way, you retain the maximum amount of information.

285-287              Were any of these variables log-transformed as is often necessary in comparative studies? If yes, please say so. If not, maybe state that the models performed well using raw data, and no transformation was necessary. It is well possible that log-transformation in the small number of closely related species made no real difference.

295        Why was “species” included as a predictor in a single-species analysis?

301        It is unclear to me how a PCA of hook length and area would capture hook shape, especially with >99% explained by PC1 (which is typically interpreted as the "size" axis), and no variation along PC2 (usually considered the first "shape" axis, i.e. variation after removing size differences). Both traits are highly correlated, and for the most part measure the same thing across the abnormal and normal sperm, in that hook area directly depends on hook length, which mostly indicates whether there is a sizeable hook in the first place. So, you could have used either of these two traits and found just about the same result, namely having a hook is different from not having one (or only a stump). Overall, then, there really is not much gained from this PCA.

322-328              I guess these results came out of the same analyses as those of the next paragraph (i.e. intra- and interspecific variation based on your LM with species as a factor)? If so, please specify. Then also, did you test for interactions between your predictors and species, thus allowing for slopes to differ between species? Finally, if these results indeed stem from an LM with species as a grouping factor, it seems a bit odd to talk about regression coefficients, because regressions do not include categorical variables. Or did you really run regressions without such grouping? If so, this would be wrong because data within species are not independent.

389        (and further occurrences) It is not clear to me what the authors mean be F(3,2). I expected all these variables to be continuous, in which case the df per term should be 1, not 3. A df=3 implies a categorical variable with 4 factor levels.

403        Why here suddenly p < 0.05 instead of the precise p-values as everywhere else? Then also, I honestly doubt the reliability of any model that leaves a single residual degree of freedom after accounting for numerous predictors in the model. I do not think you have the statistical power to draw any reliable conclusions. Also, did you check for collinearity among predictors (given they were correlated with one another)?

410        Please explain what the numbers following lambda mean. I assume these are the p-values associated with lambda=0 and =1? But in which order? I find it a bit curious that all lambda-values were precisely 1.000, but then sometimes P=1 for L=0 and sometimes for L=1. I would imagine, with L=1.000, all P for L=1 should be P=1, and all P for L=0 should be low, if not significant. Please double-check.

421-424              Please check tense.

476-478              The authors set this paragraph up in a way that suggests they conducted an intraspecific study. They list some comparative studies in support of some association between postcopulatory sexual selection and sperm hooks, then find it interesting that these studies were interspecific whereas two intraspecific studies on the house mouse found no support (also see my comment on this regarding the introduction). Finally, they state that their interspecific comparison corroborates these intraspecific results. I think this line of argumentation should be drafted more clearly and critically of the severe limitations of these analyses.

501        Please check singular/plural in “an abnormal hook phenotypes”

Reviewer 2 Report

This manuscript uses electron microscopy and video microscopy to test hypotheses on the relationship of sperm hooks on sperm performance, specifically, motility and aggregation of sperm. The manuscript is overall very well written, the introduction is clear and informative, the methods are (almost burdensomely) thorough, the statistical treatments are appropriate, the results displayed appropriately, and the discussion is interesting. Exactly as I was thinking of criticizing the in vitro methods and how that may explain the apparent lack of adaptive value to sperm hooks, I reached the conclusion and the authors discussed this exact issue. This paper is excellent and I have no reservations recommending publication. I only have one minor suggestion, which the authors may ignore if they wish, to improve the paper. Figure 4 is the only one of sperm and sperm hooks, and is only from one species, and mostly focusing on abnormal hooks, with little context for ‘normal’ sperm hooks, and certainly not from several species. The audience of this journal may not be familiar with the shape and appearance of sperm hooks, the manuscript would greatly benefit from an additional figure showing sperm (with flagella) and sperm hooks from multiple species.

Author Response

We greatly appreciate the kind review of our manuscript and have followed the suggestion to include images from all focal species with normal hooks, as well as a full sperm image including a flagellum. Please see the new Figure 1.

Round 2

Reviewer 1 Report

I thank the author for a thorough and clear revision. However, there are just a few details or clarifications from my end where there may have been some misunderstanding.

Hierarchical model

My apologies, I misspelt. What I meant was a model with species > male > sperm, meaning that male ID would be nested within species ID, individual sperm within males, and replicate measurement within sperm while of course using raw measurements (now ignoring family, although this could obviously be nested between species and male). This way you could retain the full information available instead of throwing some of it away by averaging within sperm and from there within males (i.e. losing variation at each level). It may not make a big difference and I leave this at the discretion of the authors, but in principle it would better reflect the design and structure of the data.

F statistics

Thank you for clarifying that all predictors were indeed continuous. However, I guess what I got confused by (and probably won’t be the only one), is that you say, for example on line 708, “sperm hook width significantly associates with both the frequency…and size…”, but then list the F statistics and p-values of the two full regression models (based on degrees of freedom). This is confusing, in that the statistical information is dissociated form the statement it is expected to support, which here would be the partial correlation between sperm hook width and either aggregation frequency or size. Since these are already reported in the table, you could easily just refer to them (and omit these full model statistics) or otherwise be clear about significance of the model vs. significant effect of a given predictor.

This issue gets even more pronounced toward the end of the paragraph where you report these full model statistics with up to 3 or 4 effect dfs while mentioning only a subset of included predictors.

PGLS

Thank you for your explanation of lambda. However, there must have been some misunderstanding, maybe it was because I wrote L instead of lambda. My point was not about the values of lambda themselves, but the associated p-values (if these are the values in parentheses in Table 2). The point is that the PGLS estimates the lambda value by maximum likelihood and then conducts two comparisons in likelihood ratio tests, one of the model with estimated lambda (= ML model) against a model with lambda fixed at 0 and one against a model with lambda fixed at 1. For each comparison, the PGLS function reports a p-value (and also 95%CI) to essentially indicate whether or not the estimated lambda deviates from these bounds.

My first concern is that it is not specified in the paper in which order the above two p-values are reported. This matters, because a significant difference of the ML model from lambda = 0 is very different from one with lambda fixed at 1. The first suggests that lambda is different from the null expectation of lambda=0, but in the second, lambda would be significantly lower than the null expectation of lambda=1. Therefore, it is important to know which p-value is associated with which fixed lambda value.

Now, I assume that the values are reported in the order provided by the PGLS function (i.e. first test against lambda=0, then against lambda=1). It is perfectly fine to obtain an estimated lambda=1.000, meaning it is precisely at its upper bound. But I find it extremely curious that the lambda value of the second model in Table 2 (and several thereafter) is reported to be lambda = 1.000 (1, 0.01) or similar. Here, (1, 0.01) implies that the model with the estimated lambda of 1.000 deviates significantly (P=0.01) from a model with lambda fixed at 1 but is no different from a model with lambda=0 (P=1). This would mean that the lambda at its upper bound deviates from this very same bound while being close to its opposite (lower) bound. This seems confusing, and the problem is not fixed by considering the order of the p-values reversed (i.e. against lambda=1, then lambda=0). Rather, it appears that the order could be inconsistent. I have never seen this and simply wanted to encourage the authors to please verify these values (e.g. order occasionally switched in reporting?) and also to explain to readers somewhere around that table what these values in parentheses stand for. If the results accurately reflect the PGLS output, maybe it is all a matter of extremely small sample size that simply makes it impossible to estimate lambda properly.

Then I also have another few specific points to consider:

32          This first sentence reads as if evolutionary diversification was a consequence of sperm being the most diverse cells, but it’s probably rather the opposite (i.e. diversification was so extreme that sperm became the most diverse cells). In principle, you could also just keep one or the other.

463        Strictly speaking, the phylogeny is incorporated as a covariance matrix in the error structure of the generalized least-squares model. Covariate typically refers to a predictor variable.

542        I don’t mind you conducting a PCA, but my main concern in my previous review was the term “shape”, because shape is the variance left after removing the size effect. Both area and length essential measure hook size, not shape, so “sperm hook size” would be a more appropriate description than “sperm hook shape” for PC1. PC2 would then measure hook shape (i.e. after removing size variance), and it contributes virtually no variation, indicating that hooks don’t really differ in shape, but mostly in size.

567        If you say that regression coefficients indicated correlations between traits, it would help to report these coefficients instead of F statistics. That said, regarding terminology, talking about and reporting “correlation coefficients” would be even better conceptually because ultimately you want to test for correlations between these traits (i.e. how strongly do X and Y covary?) and not regressions (how does X predict Y?). For statistics, t (with sign) would then also be better than F, even though both are obviously intimately related.

624        This would now be Figure 2A (also update remaining references to this figure)

771ff     Maybe also here consider reporting statistics more suitable for correlations rather than regressions

882        yeilded -> yielded
